# Fixing Implicit Derivatives: Trust-Region Based Learning of Continuous Energy Functions

**Matteo Toso**
CVSSP,
University of Surrey

**Neill D. F. Campbell**
University of Bath

**Chris Russell**
CVSSP, University of Surrey
and The Alan Turing Institute

## Abstract

We present a new technique for the learning of continuous energy functions that we refer to as **Wibergian Learning**. One common approach to inverse problems is to cast them as an energy minimisation problem, where the minimum cost solution found is used as an estimator of hidden parameters. Our new approach formally characterises the dependency between weights that control the shape of the energy function, and the location of minima, by describing minima as fixed points of optimisation methods. This allows for the use of gradient-based end-to-end training to integrate deep-learning and the classical inverse problem methods. We show how our approach can be applied to obtain state-of-the-art results in the diverse applications of tracker fusion and multiview 3D reconstruction.

## 1   Introduction

Although deep networks are now ubiquitous in machine learning and computer vision, prior to this, there was a strong and established literature on inverse problems that made use of geometric and probabilistic models to reason about the world. These models remain much easier for humans to interpret than deep learning approaches, as they perform a small number of complex operations while deep learning involves many simple operations. This is further aided by the geometric nature of many of the models that makes them easier to visualise.

Continuous inverse problems - like SfM, tomography, and EEG signal analysis - often have a strong theoretical basis, being built upon explicit Bayesian models of physical phenomena. These methods have strong and robust performances, thanks to decades of refinement, experience, and hand-crafted priors. However, they have fallen out of favour to deep-learning methods that allow for end-to-end fitting of parameters that tailor the model to directly maximise performance over a specified loss.

**Coupling parameters and solutions:**   Arguably the primary factor in their decline, is that inverse problems are much harder to train owing to the numerical optimisation that lies at the core of their use. Internally, the solution to an inverse problem is generated by fitting a model to a single instance of data as part of a search for a Maximum a Posteriori estimate. This step creates an extra layer of indirection between the solution found and the parameters that define the probabilistic model.

Without a direct coupling between the solution and the parameters it is not possible to perform end-to-end learning and these classical methods have been relegated to ad hoc post-processing of deep learning results. By formally characterising the dependency between these parameters and the resulting MAP estimates, we show how end-to-end training can integrate deep learning and classical inverse problem methods, giving rise to high-performant and interpretable models.

**End-to-end learning with inverse problems:**   This work shows how to integrate continuous energy minimisation in a general learning framework, combining models developed over decades and deep learning methods, and jointly training them end-to-end. We provide a straightforward derivation for

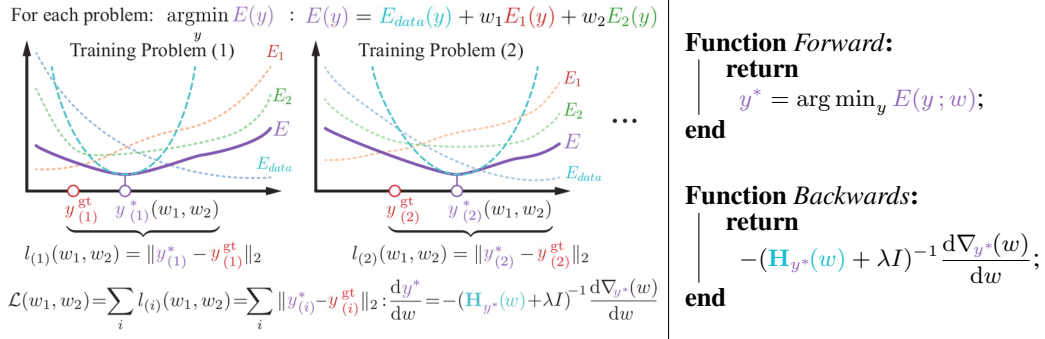

Figure 1: Predicting how the minimum moves with the parameters. Given a smooth function $E(y\,;w)$ of $y$ parameterised by $w$ (purple curve), we can locally approximate it by a quadratic function (cyan curve) about a minimum $y^*$. The quadratic approximation has an analytic minimum; by predicting how it varies with small changes in parameter, we predict how the location of the minimum of $E$ changes (this step is equivalent to performing an iteration of a trust-region method). This gives rise to the forward/backwards components. $\lambda$ is fixed to be $0.1$.

the derivatives of minima of smooth energy functions with respect to a set of general parameters that control the shape of the energy function.

To this end, we formulate the location of a local minimum as a function of the parameters controlling the energy function. This is reminiscent of the Wiberg optimisation, in which the variables are partitioned into two sets with one replaced by an analytic formula for its minimum. As such, we refer to our novel technique as *Wibergian Learning*.

**Our contribution:** We propose the first stable method that allows us to perform end-to-end discriminative learning for the location of minima on arbitrary continuous energy functions. We do this by characterising the minimum $y^*$ as a local function of the controlling weights $w$ of the energy function. From this we derive $dy^*/dw$ which allows us to train the energy function discriminatively with standard stochastic gradient descent methods used in deep learning.

We test our approach on two standard problems from computer vision and approached as energy optimisation tasks in the past. The first is tracker fusion, formulated as the minimisation of a robust non-convex loss, and the second is human pose estimation, approached as a problem of non-rigid structure from motion using a mixture of pre-trained bases.

Samuel *et al.* [1] took a similar approach to ours to train continuous Markov Random Field Models. However, for non-quadratic functions their method suffers from exploding gradients and does not converge if the determinant of the Hessian tends to zero about local minima. To avoid this instability, we make two contributions. We first offer a novel derivation with same result as [1], but by using fixed-points of the Newton-step algorithm rather than implicit functions. This, then, allows us to replace the Newton-step algorithm with a more stable-trust region algorithm which preserves the same fixed-points, where Newton-step converges, but is guaranteed to converge in a wider range of optimisation problems. Moreover, this modification bounds the size of the gradient update, explicitly preventing exploding gradients from being caused by a single component.

## 2 Related Work

The Wiberg algorithm for matrix factorisation [2, 3] was a principal inspiration for our work. This efficient optimisation technique converges to good solutions even with random initialisation. The idea underlying [2] is that by conditioning on a subset of parameters, we can make some nonlinear least squares problems linear with respect to the remaining parameters. These parameters then admit a closed form solution. By replacing the linear parameters with their analytical solution, we can arrive at an optimisation problem with fewer parameters and more informative gradients and second-derivatives. For an extensive evaluation of Wiberg-like approaches see [4], or [5] for an extension of the approach to arbitrary functions and nested variants.

Structured learning [6, 7, 8], (for an exhaustive review see [9]) concerns itself with learning the optimal energy functions to maximise a predefined criterion over the training set. While we deal with variables $y$ with continuous states, structured learning considers an $y$ that takes one of a (typically exponentially large) set of discrete states, and learns a function that ideally separates the cost of a particular pre-given labelling from all other possible labellings. As such, unlike our method, structured learning requires first discretising the problem space.

Meta-learning, also combines two optimisation processes: a learner that improves on a given task, and a meta-learner that optimises the learner. Popular applications include *(i)* "learning optimisers" using meta-learning to determine: the learner's update rules [10]; the priors and regularisers of a non-linear least squares optimiser [11]; the learning rate of a neural network [12]; or to reduce overfitting [13]. *(ii)* "few-shot learning" e.g. [14, 15], and *(iii)* "hyper-parameter optimisation" which has been characterised as meta-learning for a single task [16]. Many works in *(iii)* have focused on efficient characterisation of the inner learner. [17] suggested reconstructing the learner's optimisation trajectory rather than storing it; [18] focused on quadratic training criterion and showed Cholesky decomposition can be used to compute the gradient with respect to hyper-parameters. Finally, [16] characterised the inner optimisation procedure as a dynamical system. In contrast, our approach, and [1] directly describe the behaviour of a function's local minimum.

Finally, others have developed methods to integrate optimisation processes and neural networks by treating optimisation process as custom nodes within the network: [19] formulated quadratic programs solvers as a single, differentiable component, while [20] generalised back-propagation to matrix operations, to produce custom components for structured matrix computation including second order pooling. [21] propagated gradients through an existing least squares optimisation, but did not attempt to modify the optimisation itself.

## 3 Formulation

Let $E(y\,;w,x)$ be an energy, corresponding to the negative logarithm of an (unnormalised) probability function, defined over variables $y$, and parameterised by weights $w$ and pre-existing data $x$. It is traditional to perform optimisation to find the Maximum a Posteriori or (MAP) solution defined as

$$y^* = \arg\min_y E(y\,;w,x). \tag{1}$$

The framework is based on three assumptions: *(i)* The true distribution that generated the data is actually contained in the family of models described by the parameterised energy; *(ii)* the optimal parameters $w$ corresponding to that generating distribution can be determined; and *(iii)* the MAP solution coincides with the desired output, for example the one that minimises an expected loss. In practice, none of these assumptions are guaranteed to hold [22].

**Decoupling assumptions:** To make progress, we decouple these assumptions, as is common in discriminative approaches. We treat $E(y;w,x)$ as an arbitrary cost function, instead of the log-probability of a distribution, and choose $w$ so that $y^*$ is optimal with respect to some pre-existing loss function $\ell(\cdot)$ defined over the empiric distribution of training data. That is, we seek $w$ such that

$$\arg\min_w \sum_{x \in X} \ell(y^*(w);x) : y^*(w) = \arg\min_y E(y\,;w,x). \tag{2}$$

What makes this challenging is the decoupling of the losses on the two sides of the equation; the ideal value of $y^*$ that minimises the loss $\ell(\cdot)$ will not be the minimiser of the energy $E(\cdot)$.

**Local reparameterisation:** In light of this, we propose a novel local reparameterisation of $y^*$ as a function of $w$, *i.e.* $y^*(w)$, and show that this allows us to compute $dy^*/dw$ enabling the efficient learning of $w$ using standard methods for stochastic gradient descent and as part of an end-to-end learning framework. The key insight to our approach is that if $E(\cdot)$ is sufficiently smooth and well-behaved, the change in the solution $y^*(w) \to y^*(w')$ caused by a small perturbation of $w \to w'$ is well approximated by a single step of either Newton method, or a more robust alternative, on $y$ on the new function $E(y\,;w',x)$, starting from the current solution at $y^*$.

We begin by presenting our result in the general case, before discussing its application in end-to-end learning and detailing efficient solutions for two important special cases in the supp. materials.

**The General Case**   Given a local minimum $y^*$ of the following equation

$$y^*(w) = \arg\min_y E(y\,;w), \tag{3}$$

we are interested in characterising how the location of the local minimum varies with change in $w$. We drop the dependency on $x$ for clarity of notation. We present an informal derivation assuming that the procedure works, and leave the formal derivation to the supplementary material. We assume the local neighbourhood about $y^*(w)$ is strongly convex; and that Newton's method given $y^*(w)$ as an input will converge in a single iteration.

**Newton update w.r.t. $y$:**   Considering the second-order Taylor expansion of the energy around $y$, we have

$$E(y'\,;w) \approx E(y\,;w) + (y'-y)[\nabla E(y\,;w)] + \tfrac{1}{2}\,(y'-y)^{\mathrm{T}}[\mathbf{H}E(y\,;w)](y'-y), \tag{4}$$

where $\nabla E(y\,;w)$ denotes the Jacobian and $\mathbf{H}E(y\,;w)$ the Hessian of $E(\cdot)$ with respect to $y$ evaluated at $y$. If we are sufficiently close to a minimum of $E(\cdot)$, the expansion well models the function and the minimum of the two coincide. This leads to Newton's update rule

$$\arg\min_{y'} E(y'\,;w) \approx y - [\mathbf{H}E(y\,;w)]^{-1}\nabla E(y\,;w). \tag{5}$$

If we evaluate this at the minimum $y = y^*(w)$, we have

$$y^*(w) = y^*(w) - [\mathbf{H}E(y^*(w)\,;w)]^{-1}\nabla E(y^*(w)\,;w), \tag{6}$$

with $\nabla E(y^*(w)\,;w) = \mathbf{0}$ at optimality.

**Updating parameters $w$:**   We then ask, "What would a single iteration of Newton's method do, if the parameters $w$ are updated?" The answer is that, for sufficiently small updates of $w$, $y^*(w)$ remains in the strongly convex region about the new minimum and one iteration of Newton's method moves $y^*(w)$ directly towards the new minimum $y^*(w')$. Writing $w' = w + \Delta$, then as $\Delta \to 0$ only a single iteration of Newton's method is needed to get arbitrarily close to the new minimum. That is,

$$y^*(w+\Delta) \approx y^*(w) - [\mathbf{H}E(y^*(w);w+\Delta)]^{-1}\nabla E(y^*(w);w+\Delta). \tag{7}$$

Writing $\mathbf{H}_{y^*}(w)$ as shorthand for the Hessian of $E(y\,;w)$ w.r.t. $y$ at the fixed location $y^*$, *i.e.* $\mathbf{H}E(y^*;w)$, and $\nabla_{y^*}(w)$ as shorthand for the Jacobian of $E(y\,;w)$ with respect to $y$ at $y^*$, *i.e.* $\nabla E(y^*;w)$, we rearrange and normalise the above equation to get

$$\frac{y^*(w+\Delta)-y^*(w)}{\Delta} \approx -\frac{\mathbf{H}_{y^*}^{-1}(w+\Delta)\nabla_{y^*}(w+\Delta)}{\Delta} = -\frac{\mathbf{H}_{y^*}^{-1}(w+\Delta)\nabla_{y^*}(w+\Delta)}{\Delta} + \frac{\mathbf{H}_{y^*}^{-1}(w)\nabla_{y^*}(w)}{\Delta} \tag{8}$$

with the final expression following from the fact that $\nabla E(y^*(w);w) = \mathbf{0}$. Proof that this expression converges as $\Delta \to 0$ is given in the supplementary materials.

In the limit $\Delta \to 0$, and again using $\nabla_{y^*}(w) = \mathbf{0}$ by definition, this allows us to derive

$$\frac{\mathrm{d}y^*}{\mathrm{d}w} = -\frac{\mathrm{d}\mathbf{H}_{y^*}^{-1}(w)\nabla_{y^*}(w)}{\mathrm{d}w} = -\frac{\partial\mathbf{H}_{y^*}^{-1}(w)}{\partial w}\nabla_{y^*}(w) - \mathbf{H}_{y^*}^{-1}(w)\frac{\partial\nabla_{y^*}(w)}{\partial w} = -\mathbf{H}_{y^*}^{-1}(w)\frac{\partial\nabla_{y^*}(w)}{\partial w}, \tag{9}$$

Here we use the partial derivative to emphasise that the value of $y^*$ comes from a previous iteration of Newton's method, and that it does not vary with $w$.

**Trust-Region Based Robustness**

The derivation up to this point gives the same update step as Samuel *et al.* [1]. However, it is immediately apparent that if the function descends sharply into a flat region about a local minimum, and is better approximated by a higher-order function than by a quadratic function, the Hessian may tend to zero around the minimum, and $\mathbf{H}_{y^*}^{-1}(w)$ is ill-defined. Even if $\mathbf{H}_{y^*}(w)$ is non-zero, it may become arbitrarily small leading to exploding gradients. To avoid such issues, we turn to the trust-region method of [23], which replaces $\mathbf{H}_{y^*}(w)$ in the update step of Newton's method with

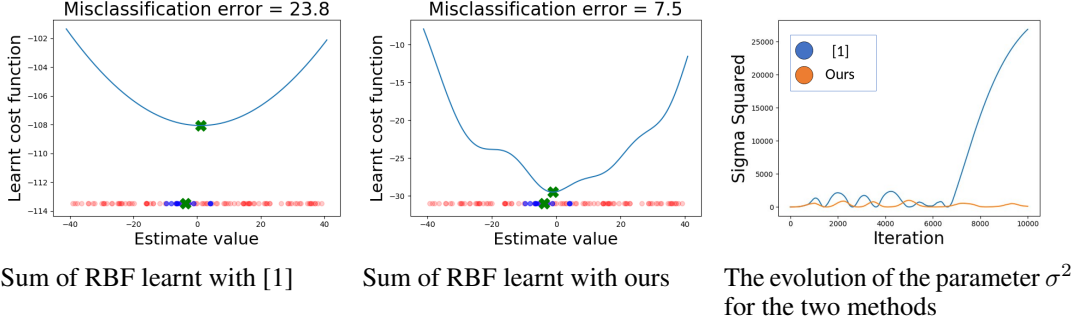

Figure 2: Learning a RanSac like function. A simple illustration of learning the kernel width $\sigma^2$ on 1D RBF functions so that the minimal cost solution corresponds to the mean of a dense set of 10 inliers (blue samples in the first two graphs) and discards 100 outliers(red samples). The green cross indicates the function minima corresponding to the estimate found by each method.

$\mathbf{H}_{y^*}(w) + \lambda I$ for some $\lambda \geq 0$, where $I$ is the identity matrix. As shown by [23], this is equivalent to minimising the same quadratic problem at each iteration of Newton's method subject to additional constraints on the maximal size of the step, and it inherits the same quadratic convergence in the local neighbourhood of a strongly convex minimum, with stronger convergence properties elsewhere. Use of the trust-region instead of Newton's method gives rise to the damped gradient prediction

$$\frac{\mathrm{d}y^{*\,(d)}}{\mathrm{d}w} = -\left(\mathbf{H}_{y^*}(w) + \lambda I\right)^{-1} \frac{\partial \nabla_{y^*}(w)}{\partial w} . \tag{10}$$

Using the argument of [23], this value can be interpreted as damped variant of the original gradient in the following sense: while $\frac{\mathrm{d}y^*}{\mathrm{d}w}$ is the solution to the quadratic programme $\arg\min_y ||\mathbf{H}_{y^*}(w)y - \frac{\partial \nabla_{y^*}(w)}{\partial w}||_2^2$, $\frac{\partial y^{*\,(d)}}{\partial w}$ is the solution to the same programme subject to the requirement that $||y||_2^2 \leq k$ for some $k$.

Compared to the undamped formulation of (9), trust-region methods converge to a true minimum for a strictly larger class of functions making the new approach directly applicable to a wider range of problems. Moreover, we can bound the magnitude of the damped gradient directly, in all circumstances. As $\mathbf{H}_{y^*}(w)$ is positive semi-definite, and $\lambda I$ positive definite, we have $||\frac{\mathrm{d}y^{*\,(d)}}{\mathrm{d}w}|| \leq \lambda^{-1}||\frac{\partial \nabla_{y^*}(w)}{\partial w}||$, and exploding gradients can no longer be created by a single layer.

This is not just a convenience: in section 4, we demonstrate a problem that fails to converge using [1]; but gives state of the art results using our formulation. In practice no tuning or online adaption of the value of $\lambda$ is needed, and we simply set it to $0.1$.

If the energy is quadratic, trust region analysis is unneeded, however [1] may still fail to converge for ill-posed quadratic problems. Analysis showing closed form updates for well- and ill-posed quadratic energies guaranteed to converge are in the supplementary materials.

**Using the Derivative in Learning** The derivatives we have specified are quite general, and, importantly, they make no assumptions about the energy minimisation technique used to obtain the optimum. In practice, approximate second order approaches such as L-BFGS [24, 25] converge to a neighbourhood about the minimum, but the solution found does not satisfy the fixed point equation (6). In this case, a single step of (trust-region) Newton's method is required for the numeric gradients and the analytic solution found to coincide. This instability primarily affects the numeric derivatives of the analytic solution described in equation (9). The right-hand term, set to $0$ a priori, describes the drift in Newton's method due to not starting at an exact local minimum.

Given knowledge of how to compute the gradients, energy minimisation can be treated as a component of any end-to-end training network, which makes use of stochastic subgradient descent, and integrated directly. We exploit this in both our examples. In tracking, we have simple stub functions that weight our confidence in particular bounding-boxes based on how their scale has changed from the first frame. In 3D reconstruction, we take a weighted average of multiple reconstructions and jointly learn these weights and the ideal energy function.

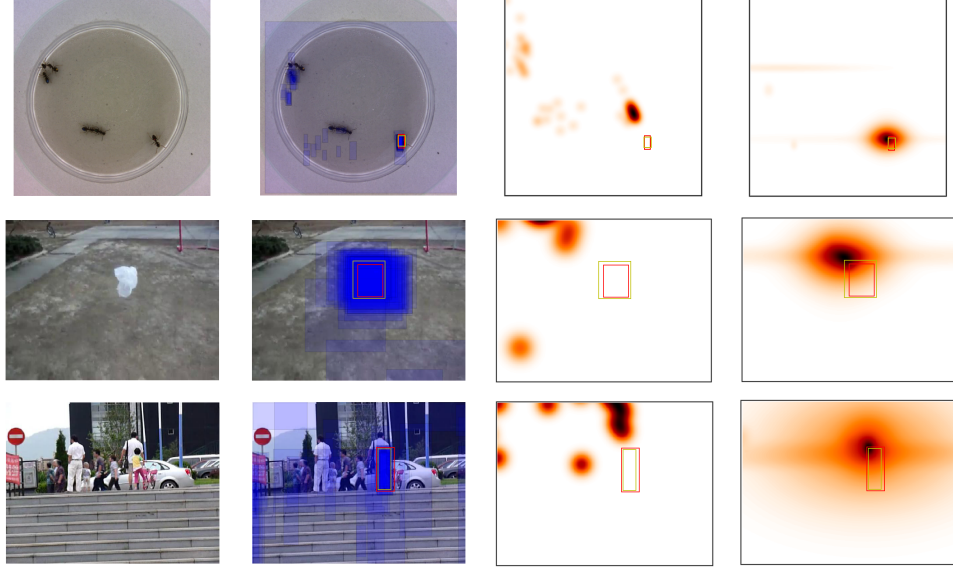

Figure 3: Learning a better energy function for tracker fusion. **Left:** Input image. **Centre Left:** Overlay of 72 tracker boxes (blue), ground-truth detection (red) and our prediction (yellow). **Centre Right:** The initial energy used to predicting the top left corner of the box locations before training. **Right:** The energy used to predict the same corner of box locations after training (red continues to indicate ground-truth). Our learning mechanism naturally adapts the form of the energy so that it contains fewer poor local minima that the optimisation could stick in, and directs the local minima to a better location.

**RanSaC as an Illustrative Example**  We demonstrate our approach on a simple 1-dimensional example. We consider the problem of estimating the mean of a set of 10 inliers sampled from a normal distribution $N(U[-40, 40], 4^2)$ in the presence of 100 outliers drawn from a broad uniform distribution $U[-40, 40]$. This can be formulated as an MLESaC[26] type optimisation where the mean is estimated by minimising a one-dimensional sum of $RBF$ functions centred on the samples i.e.: $\hat{\mu} = \arg\min_t E(t, x; \sigma) = \sum_i -\exp(-(x_i - t)^2/\sigma^2)$. We compare our approach, and that of [1] to find optimal value of $\sigma$ to minimise the squared error between the estimated mean and its true value. The large amount of volatility in this problem means that adaptive gradient methods[27, 28] stop early and do not converge to optimal solutions. Instead we use stochastic gradient descent with a strong momentum value of $0.999$ to damp the oscillations. For any choice of step-size and momentum, with probability 1 we will eventually draw a set of points that have a sufficiently small curvature about the minimum, causing an arbitrarily large step for the undamped update of [1] while our update remains bounded. This behaviour can be seen in figure 2.

## 4   Tracker Fusion

We demonstrate how our approach can be used to train a model to fuse existing candidate trackers. This demonstration leverages the comprehensive evaluation work performed by the Visual Object Tracking challenge team [29]. Alongside annotated ground-truth results, the VOT team have archived the tracking results of 72 entries to their competition, and we take these results as initial candidates for tracking. The VOT challenge has been set up to evaluate several different types of tracking, both long and short-term, and with and without reinitialisation of the tracker after failure. To demonstrate the merits of our approach, we focus on short-term tracking performed without reinitialisation of the tracker. Our justification is that (i) short-term tracking is a more established benchmark with more trackers available, and that (ii) the unsupervised case, where no reinitialisation is performed, makes it easier to work with existing trackers. We take the output of all trackers run once through the sequence without intervention and learn to fuse the tracker results on any given frame. Our task is to predict the four corners of a bounding box, given a set of candidate locations from the other trackers.

Table 1: The VOT2018 challenge; showing the best 5 methods of the 72 entries on different test sets. We evaluate on two data partitions one where individual frames are randomly assigned to training and test, and another where entire sequences are assigned to each set. * = did not converge.

| Frames Assigned at Random | | Sequences Assigned at Random | |
|---|---|---|---|
| Tracker | IoU | Tracker | IoU |
| DLSTpp | 0.530 | SA_Siam_R | 0.4643 |
| FSAN | 0.490 | MBSiam | 0.4624 |
| SiamRPN [30] | 0.484 | SiamRPN [30] | 0.4621 |
| LSART[31] | 0.472 | FSAN | 0.4618 |
| R_MCPF | 0.465 | LADCF[32] | 0.4601 |
| Mean Fusion | 0.238 | Mean Fusion | 0.2455 |
| Median Fusion | 0.428 | Median Fusion | 0.4458 |
| Samuel *et al.*[1] | N/A * | Samuel *et al.*[1] | N/A * |
| Our Fusion | 0.565 | Our Fusion | 0.4960 |

To aid visualisation of the learning, we split the energy function describing the corners into two independent components, one used to predict the top left corner of the box (see Figure 3), and the other to predict the bottom right corner. Each component is modelled as a sum of 72 Radial Basis Functions (RBF), one for each prediction given by an existing tracker. The standard deviation of each RBF, along with temporal based importance weights, are learnt for each tracker, allowing us to discount trackers that are more prone to drift later on in the sequence, along scale factors that correct for trackers that consistently underestimate the size of the bounding box. These scale factors account for much of the movement of minima in figure 3. The full formulation is given in the supp. materials.

We take the centre of each RBF as a candidate solution, and use the lowest energy candidate as an initialisation for continuous optimisation of the objective using L-BFGS [24, 25]. Importantly, this optimisation is not guaranteed to find the global optimum, and our learning framework does not require it to. We simply require that, given a similar characterisations of the loss, the solution found is a local minimum that remains stable in expectation[1], and will drive the solution we find towards something that will minimise our training loss.

We take the training loss as the $\ell_1$ difference between the found minima and ground-truth solution. In practice, the method of [1] frequently failed converge on this problem. Although this could sometimes be avoided by a mixture of early stopping and a careful tuning of step sizes, this hurt the quality of solutions found and still intermittently failed.

**Results**    Our fused tracker shows state of the art results on the highly competitive VOT2018 challenge. For evaluation purposes we divide the VOT2018 dataset roughly into two thirds training/one third test by number of frames, so that no sequence occurs in both halves, training on the first half of the frames and reporting loss on the second; we also report on "missing at random frames", where frames from the same sequence occur in both training and test. Mean and median fusion are baseline "wisdom of the crowd" methods showing the effectiveness of predicting using either the mean or median prediction over the 72 methods for the bounding box corners. In Table 1 we report the intersection over union measure both for our approach, and for the top five methods from the VOT2018 challenge on the same partitions. For the newer and uncited methods, please see the appendix of [29] for further details. Results differ from [29] due to training and test partitions.

## 5    Human Pose Estimation

We further demonstrate our approach on 3D Human Pose Estimation from 2D detections. We consider both the monocular model by Tome *et al.* [21] and its multi-camera extension [33], and use our learning method to improve on their hand-tuned energy functions. We emphasises that we do not alter the optimisation used, but instead alter the shape of the energy function so that its minimum is closer to the ground-truth.

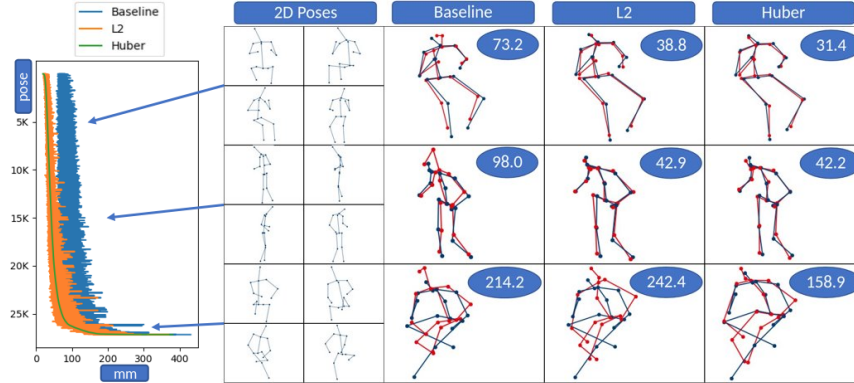

Figure 4: Reconstructed poses from our three multi-camera approaches (red) vs ground truth (blue).

**The Model**   The method [21] was based on classical basis shape approaches to Non-Rigid Structure from Motion [34] and found the reconstructed 3D pose $P$ by solving the least squares problem:

$$a^*, R^* = \arg\min_{a,R} \left\| X - s \, \Pi \, K \, R \, [\, \mu + a \cdot \mathbf{e} \,] \right\| + \left\| \sigma \cdot a \right\|_2^2 \tag{11}$$

$$P_i(R^*, a^*) = R^* \, [\, \mu_i + a^* \cdot \mathbf{e}_i \,] \tag{12}$$

where $R$ is an in-plane rotation, $a$ the basis coefficients vector, $X$ is an input $2D$ pose, $\mu$ a rest shape, $\mathbf{e}$ a tensor of basis vectors, $\Pi$ the orthographic projection matrix, $K$ a known external camera calibration matrix, and $s$ the estimated per-frame scale.

The authors also extended this approach to a mixture of models $[\mu_i, \mathbf{e}_i, \sigma_i]$, selecting the model with the minimal reconstruction error as the true reconstruction. The approach was extended to a multi-camera setting that also replaced the search over all rotations with a weighted average [33].

**Learning Parameters**   The optimisation process solves a well-posed quadratic function, allowing direct computation of the gradient of $a$. Each reconstruction can be written as a direct smooth function of $a_i$ - making the full process differentiable and allowing end-to-end training.

We replaced the Huber loss used in  [33] with its smooth variant to make its transition from $\ell_2$ to $\ell_1$ optimisable, and following the argument in [33] replaced the choice of best fitting model with a weighted average. sto speed up training, and would not be required: choosing the best model and updating it induces a valid subgradient.

Where possible, parameters were initialised to the previously used values. New terms that are multiplied by or added to an existing function were respectively set to $1$ or $0$. The model parameters were then trained and tested using the stacked hourglass detections of Martinez *et al.*[35]. This change to more reliable detectors, but with a different distribution of errors, leads to the large increase in reconstruction error between [21, 33] and our baselines.

**Results**   We evaluate our approach on the Human3.6M dataset[40], over the standard training and testing sets for both monocular and multiview reconstruction. Results can be seen in Table 2. In monocular reconstruction, we show a 12mm reduction in error vs. [21], whose pose model was generated with probabilistic PCA and hand-tuned the remaining weights. However, performance remains worse than methods such as [35].

We have greater success in the multiview case. Training the humane pose model and camera weights reduces the error by 34mm with respect to the baseline, and by jointly optimising these parameters alongside those that were originally found by grid-search (see table 3) we obtain state-of-the-art results. This corresponds to a 6mm reduction in error over the hand-tuned approach [33]. The closest competitor [38] made use of a personalised model of 2D joint appearance while we learn a better model for fitting 3D pose; as such, the two approaches are strongly complementary.

**Computational efficiency:**   Our approach, being a gradient-based online method, is much more scalabale to large datasets and optimisation over large numbers of parameter than grid search. Given

Table 2: Average per joint 3D reconstruction error on Human3.6M, expressed in mm.

| Monocular | Dir. | Disc. | Eat | Greet | Phone | Photo | Pose | Purch. | Sit | SitD. | Smoke | Wait | WalkD. | Walk | WalkT. | **Avg** |
|---|---|---|---|---|---|---|---|---|---|---|---|---|---|---|---|---|
| Tome *et al.* [21] | 65.0 | 73.5 | 76.8 | 86.4 | 86.3 | 110.7 | 68.9 | 74.8 | 110.2 | 173.9 | 85.0 | 85.8 | 86.3 | 71.4 | 73.1 | 88.4 |
| Martinez *et al.* [35] | 51.8 | 56.2 | 58.1 | 59.0 | 69.5 | 78.4 | 55.2 | 58.1 | 74.0 | 94.6 | 62.3 | 59.1 | 65.1 | 49.5 | 52.4 | 62.9 |
| Ours - Baseline | 100.8 | 104.1 | 103.1 | 108.4 | 109.2 | 122.3 | 98.1 | 130.7 | 136.6 | 192.1 | 102.0 | 108.1 | 113.6 | 98.4 | 104.9 | 114.2 |
| Ours | 57.5 | 68.5 | 65.9 | 71.3 | 82.0 | 80.2 | 57.5 | 60.7 | 102.8 | 136.7 | 75.8 | 70.6 | 74.4 | 58.2 | 64.4 | 76.1 |
| **Multicamera** | | | | | | | | | | | | | | | | |
| Trumble *et al.* [36] | 92.7 | 85.9 | 72.3 | 93.2 | 86.2 | 101.2 | 75.1 | 78.0 | 83.5 | 94.8 | 85.8 | 82.0 | 114.6 | 94.9 | 79.7 | 87.3 |
| Zhou *et al.* [37] | 54.8 | 60.7 | 58.2 | 71.4 | 62.0 | 65.5 | 53.8 | 55.6 | 75.2 | 111.6 | 64.2 | 66.1 | 51.4 | 63.2 | 55.3 | 64.9 |
| Pavlakos *et al.* [38] (a) | 41.2 | 49.2 | 42.8 | 43.4 | 55.6 | 46.9 | 40.3 | 63.7 | 97.6 | 119.9 | 52.1 | 42.7 | 51.9 | 41.8 | 39.4 | 56.9 |
| Pavlakos *et al.* [38] (b) | | | | | | | 46.0 | | 68.1 | 73.9 | | | | | | 47.8 |
| Núñez *et al.* [39] | 40.2 | 47.0 | 42.1 | 66.5 | 53.0 | 58.0 | 36.4 | 41.8 | 68.2 | 113.8 | 51.6 | 66.7 | 47.2 | **35.6** | **35.5** | 54.2 |
| Tome *et al.* [33] - L2 | 51.3 | 54.9 | 47.9 | 55.8 | 56.8 | 71.3 | 45.8 | 49.2 | 74.7 | 102.0 | 56.2 | 62.2 | 56.1 | 48.7 | 54.0 | 59.4 |
| Tome *et al.* [33] - Huber | 43.3 | 49.6 | 42.0 | 48.8 | **51.1** | 64.3 | 40.3 | 43.3 | 66.0 | 95.2 | 50.2 | 52.2 | 51.1 | 43.9 | 45.3 | 52.8 |
| Ours - Baseline | 85.7 | 90.8 | 79.8 | 87.3 | 107.1 | 94.8 | 78.5 | 87.6 | 102.0 | 100.1 | 95.2 | 85.1 | 92.3 | 85.8 | 87.4 | 91.8 |
| Ours - L2, learn only $\mu, e, \sigma, W_c$ | 50.9 | 55.0 | 49.3 | 53.7 | 70.3 | 53.9 | 48.5 | 50.0 | 63.8 | 71.9 | 60.8 | 51.5 | 57.6 | 57.7 | 57.5 | 57.8 |
| Ours - L2 | 38.5 | 44.3 | **39.2** | 42.1 | 61.9 | **44.4** | 36.0 | **38.6** | **56.7** | **65.6** | 50.6 | 41.0 | 47.7 | 45.3 | 46.6 | 47.7 |
| Ours - Huber | **38.2** | **42.2** | 39.5 | **39.1** | 57.2 | 45.2 | **34.1** | 39.1 | 57.8 | 68.0 | **48.7** | **39.1** | **46.7** | 40.5 | 41.1 | **46.1** |

Table 3: Parameters used in multi-view reconstruction. [21] and [33] optimised the parameters marked ‡ with grid-search.

| Component | Body Models | Reprojection | Cameras | Covariance | Huber ‡ | Fusion ‡ | Scaling ‡ |
|---|---|---|---|---|---|---|---|
| # Parameters | 3978 | 92 | 36 | 2523 | 9 | 6 | 4 |

$k$ parameters, $N$ possible values for each and a training set evaluation time of $t$ hours, grid search requires $t \cdot N^k$ hours. Generating the Huber-loss reconstruction over the entire multiview training set of Human3.6M takes 5 hours on a typical CPU. With $N = 10$ and the parameters of Table 3, the search would take $5 \cdot 10^{6648}$ hours, while our stochastic online approach took approximately 120 core hours to converge (i.e. much faster than searching over 2 parameters).

**Limitations:** Our approach inherits many of the advantages and disadvantages of gradient descent methods in neural nets. In particular, just as vanishing gradients, and stuck neurons are a concern, it is possible for particular components of the energy to have too narrow a range to influence the location of minima; if this is the case, they will remain fixed. As such, the use of sensible initialisations and regularisers to ensure that by default components have an initial broad range is important. These modifications typically decrease the curvature at local minima, making the use of our modified update step more important. In general, our modified update step is most important at the start of the learning process, while the final energy functions that our algorithm converges to tends to be better behaved.

## 6    Conclusion

We have presented a novel approach that allows the classical energy minimisation methods of inverse problems to benefit from the end-to-end training that has been a fundamental part of the success of deep-learning. By placing these energy minimisation techniques within the framework of end-to-end learning we have opened the door for the integration of the two approaches; allowing us to learn things that can not be easily expressed in either paradigm. For example, using the technique proposed, it is possible to use Siamese networks [41] to output confidence weights on their matches and by integrating them with standard SfM techniques, train to maximise directly the 3D accuracy of reconstructions. Equally, the routing of Capsule Networks [42] could be formulated as an energy minimisation problem (similar to the pictorial structure/spring model [43]) and jointly trained.

Our work provides a bridge that connects together variational methods and inverse problems with deep learning; and as such it is particularly well suited for modelling complex interactions that are difficult to describe with standard convolutional networks.

Code is available at: `https://github.com/MatteoT90/WibergianLearning`

**Acknowledgements:** We acknowledge funding from the RCUK Centre for the Analysis of Motion, Entertainment Research and Applications (CAMERA, EP/M023281/1) and the Royal Society.

## Footnotes

[1]In theory, this allows for the use of random restarts and non-deterministic optimisation in the energy minimisation step.

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
