[Supplementary Material · nips_supplement.pdf]

# Fixing Implicit Derivatives: Trust-Region Based Learning of Continuous Energy Functions - Supplementary Material

**Matteo Toso**
CVSSP,
University of Surrey

**Neill Campbell**
University of Bath

**Chris Russell**
CVSSP, University of Surrey
and The Alan Turing Institute

## 1 Proofs of Convergence

We consider functions that lie in $\mathbb{C}^2$ i.e. continuous functions with well-defined first ($\nabla$) and second ($H$) derivatives and focus on a class of minima for which the properties of Newton's Method are well understood. Namely, we look at strict or isolated minima $y^*$ that satisfy the following two conditions:

$$\nabla_{y^*}(w) = 0 \tag{1}$$

and

$$H_{y^*}(w) \text{ is positive definite.} \tag{2}$$

These conditions are sufficient but not necessary for isolated minima e.g. $f(y) = y^4$ has an isolated minima at $y = 0$, but $f''(0) = 0$.

We further assume that $\nabla$ and $H$ are (Lipschitz-) continuous functions of $y$, and of their controlling parameters $w$.

Drawing heavily from the book Practical Methods of Optimization[1], our proof has two parts: (i) show that one iteration of Newton step is sufficient to predict the new location of the minimum in the limit. (ii) Prove that the derivative of one iteration of Newton Step with respect to $w$ exists. The second step is not completely obvious as it relies upon the stability of $H^{-1}$, something which is fortunately guaranteed by the standard assumption in optimisation that $H$ is positive definite about the minimum.

**(i) In the limit one step of Newton's Method is sufficient to find the new minima**

We provide a sketch proof for the 1-D case, with the multi-dimensional case following directly.

Given a strict minimum satisfying constraints (1) and (2), Newton's method has a squared rate of convergence. That is to say, if we write $N(y)$ for a single iteration of Newton's method starting from $y$, there is a sufficiently small radius $r$ such that for all $y : |y - y^*| < r, |N(y) - y^*| < r^2$ (see Theorem 3.11 [1]). From this it follows that as $r \to 0$,

$$\frac{y - y^*}{y - y^*} - r^2/r \leq \frac{y - N(y)}{y - y^*} \leq \frac{y - y^*}{y - y^*} + r^2/r \tag{3}$$

Hence it follows from the fact[1] that $r \to 0$ as $\Delta \to 0$ that $\frac{y-N(y)}{y-y^*} \to 1$. Consequentially,

$$\frac{y - N(y)}{\Delta} = \frac{y - y^*}{\Delta} \tag{4}$$

and in the limit

$$\frac{dN(y)}{dw} = \frac{dy^*}{dw}, \tag{5}$$

with the multi-dimensional derivation following directly.

**(ii) The derivative of one step of Newton's Method exists**

As with the previous proof, we show this result in $\mathbb{R}$ and then generalise to $\mathbb{R}^n$.

We consider $E_w : y \to z$, a smooth function about a minimum $y^*$. We consider a strict local minimum satisfying conditions (1) and (2). Following the derivation in the main body of the paper, we have that $H_{y*}(w)$ is a continuous function of $w$, and as $H_{y^*}$ is strictly positive, there must exist an open neighbourhood about $w$ where $H_{y*}^{-1}(w)$ is smooth. As $\nabla_y(w)$ is also smooth with respect to $w$, their product is smooth and the derivative exists.

Generalising this to $\mathbb{R}^n$ is straightforward. Just as in the 1-D case, if $H_{y^*}(w)$ is strictly positive definite there is a neighbourhood about $w$ in which $H_{y^*}$ remains strictly positive definite and invertible, guaranteeing that the inverse remains smooth.

This property only holds for minima with positive definite Hessians. Saddle points and many other minima where $H_y$ is not invertible may not have well defined limits.

## 2   Common Special Cases

We now consider two special cases that have unique solutions that avoid the need to be characterised in terms of a neighbourhood about a local minimum: (i) ill-posed quadratic problems in which multiple minima may exist, and (ii) well-posed quadratic problems with a singular minimum. The second case has a direct derivation and the full rank for well-posed problems permits the use of the Cholesky decomposition. As an example of a reference implementation, we consider the "fast" variant of the Tensorflow least-squares solver (`tf.linalg.lstsq`).

The first case is more involved and we include a derivation since we were unable to find a full description in the literature. The rank deficiency requires that we use a robust decomposition ($LDL$ rather than Cholesky) and we need to exploit identities for the derivatives of the pseudo-inverse. Again, as a reference implementation, we consider the Tensorflow least-squares solver but with the "slow" variant.

**Arbitrary quadratic problems:**   For the case where $E(y\,;w)$ is an arbitrary quadratic function we have

$$E(y\,;w) = ||A_w y - b_w||_2^2 \tag{6}$$
$$= (A_w y - b_w)^{\mathrm{T}}(A_w y - b_w). \tag{7}$$

A global minimum with respect to $y$ occurs when

$$\frac{\mathrm{d}E(y\,;w)}{\mathrm{d}y} = 2(A_w y - b_w) = \mathbf{0} \tag{8}$$

and thus

$$A_w y^* = b_w \ \Rightarrow \ y^* = A_w^+ b_w, \tag{9}$$

where $A_w^+$ is the pseudo-inverse of $A_w$. Importantly, use of the pseudo-inverse makes the problem well-posed; if the solution to equation (6) is ill-determined, it returns the unique minimiser with the minimal $\ell_2$-norm. Using the derivative of the pseudo-inverse of [2] we have

$$\frac{\mathrm{d}A_w^+}{\mathrm{d}w} = -A^+ \left(\frac{\mathrm{d}A}{\mathrm{d}w}\right) A^+ + \ A^+ A^{+\mathrm{T}} \left(\frac{\mathrm{d}A^{\mathrm{T}}}{\mathrm{d}w}\right)(I - AA^+)$$
$$+ \ (I - A^+ A)\left(\frac{\mathrm{d}A^{\mathrm{T}}}{\mathrm{d}w}\right) A^{+\mathrm{T}} A^+, \tag{10}$$

and we can derive $^{\mathrm{d}y^*(w)}/_{\mathrm{d}w}$ as

$$
\begin{aligned}
\frac{\mathrm{d}y^*}{\mathrm{d}w} &= A_w^+ \frac{\mathrm{d}b_w}{\mathrm{d}w} + \frac{\mathrm{d}A_w^+}{\mathrm{d}w} b_w \\
&= A_w^+ \frac{\mathrm{d}b_w}{\mathrm{d}w} - A^+ \frac{\mathrm{d}A}{\mathrm{d}w} A^+ b_w \\
&\quad + A^+ A^{+\mathrm{T}} \frac{\mathrm{d}A^\mathrm{T}}{\mathrm{d}w} \left( I - AA^+ \right) b_w + \left( I - A^+ A \right) \frac{\mathrm{d}A^\mathrm{T}}{\mathrm{d}w} A^{+\mathrm{T}} A^+ \frac{\mathrm{d}A_w^+}{\mathrm{d}w} b_w .
\end{aligned}
\tag{11}
$$

**Well-defined quadratic problems:** In the final, and most restrictive case, we consider solutions of equation (6) where the matrix $A$ is known to be full rank, and the problem either fully- or over-determined. Here we solve the problem via Cholesky decomposition, by using the identity $y^* = (A^T A)^{-1} A^T B$, and the derivative of a matrix inverse [3] as

$$
\frac{\mathrm{d}A^{-1}}{\mathrm{d}w} = A^{-1} \frac{\mathrm{d}A}{\mathrm{d}w} A^{-1}.
\tag{12}
$$

In this case we have

$$
\frac{\mathrm{d}y^*}{\mathrm{d}w} = (A^\mathrm{T} A)^{-1} A^\mathrm{T} \frac{\mathrm{d}b}{\mathrm{d}w} + (A^\mathrm{T} A)^{-1} \frac{\mathrm{d}A^\mathrm{T}}{\mathrm{d}w} b + (A^\mathrm{T} A)^{-1} (2 \frac{\mathrm{d}A^\mathrm{T}}{\mathrm{d}w} A)(A^\mathrm{T} A)^{-1} A^\mathrm{T} b.
\tag{13}
$$

To demonstrate the effectiveness of our approach we consider two traditional problems of computer vision. Namely, tracker fusion as an energy minimisation problem using a robust non-convex loss-function and a non-rigid structure from motion approach to the 3D reconstruction of human poses using a pre-learnt basis functions.

## 3 Tracker Fusion

The energy for the top-left/bottom-right corner $[y_0, y_1]$ is modelled as a mixture of Gaussians of the form:

$$
E(y) = \sum_{t \in \mathcal{T}} m_t \exp \left( -\frac{(y_0 - a_t)^2}{\sigma_{T,t}^2} - \frac{(y_1 - b_t)^2}{\sigma_{L,t}^2} \right)
\tag{14}
$$

where the mixing weight $m_t$ for tracker $t$ is

$$
m_t = s_t \exp \left( -\frac{(w_{f,t} - w_{0,t})^2}{\sigma_{w,t}^2} - \frac{(h_{f,t} - h_{0,t})^2}{\sigma_{h,t}^2} \right) \theta_f
\tag{15}
$$

with $s_t$ being the per tracker scale, $w_{f,t}$ being the width of the bounding box of tracker $t$ in frame $f$, and $\theta_f$ is the temporal weight due exclusively to the frame

$$
\theta_f = \frac{1}{1 + A_t^2 f + B_t^2 f^2}
\tag{16}
$$

With $A_t$ and $B_t$ being arbitrary parameters squared to keep them non-negative. All free parameters $\sigma, \theta, s, A, B$ are tuned automatically using the method outlined in the paper.

## 4 Human Pose Estimation

The models were trained on CPU only, on a standard desktop machine.

The training data set is the one provided Martinez *et al.* [4], namely stacked hourglass detections from Human3.6M, sampled one every 5 frame. As the other multi-view approaches considered, we only used frames for which all four camera views were available.

The detections provided only have 16 joints, while the network proposed by Tome *et al.* [5] uses 17 joints; this was addressed by including a fake chest joint, set at the origin of the reference system and marked as occluded. This means that in all our reconstructions the chest joint is inferred using the physically-plausible basis and rest pose, and the reconstructed location of the other joints.

Figure 1: Error distribution on Human36M using our multi-camera model. Results are included for the untrained (Baseline) network and for the learnt shapes for the energy functions using both the $\ell_2$ and the Huber loss. For visual clarity, we sort the datapoints by order of increasing error for the Huber case (i.e. our most effective approach). All models perform well on typical input instances, and fail in a limited number of cases. These outliers, however, have a noticeable, negative impact on the average error. We also provide reconstruction examples from the low, medium and tail part of the distribution.

Figure 2: The learnt importance of joints. The figures show the individual importance of each joint on a model by model basis for the $x$ (left-hand figure) and $y$ (right-hand figure) components of the reprojection error in multiview reconstruction.

In Figure 1, we compare the error distribution of the initialised model ("Baseline") with the trained one, using both an energy function based on the $l2$ and one based on the Huber loss. To make the data clearer, the datapoints are arranged by error size according to the Huber model. We can see how both trained model are consistently - and abundantly - better than what is obtained using the original network parameters of Tome *et al.* [5]. Moreover, the Huber version of the network results generally more accurate than the $l2$ one. Moreover, both trained models show good performances on the typical inputs and fail in a limited number of cases.

We also include an up-scaled version of the third example showed in Figure 4 of the paper; Figure 3 represent an instance taken from the tail of the error distribution. Even if the reconstruction failed, our proposed approach is still able to produce a pose more physically plausible than the one suggested by the original method, having retrained the basis vectors and the rest poses.

We also provide additional results for the monocular test. To demonstrate that Wibergian learning allows the use of more complex models than standard parameter tuning, we trained three different networks, with a varying number of mixture components. We initialise these models following Tome *et al.* [5] and evaluate the resulting networks, giving the baseline results. For reference, we also include the results by Martinez *et al.*[4]. While [4] outperforms our results, the focus in our work is to propose a new learning mechanism that substantially outperforms current training for variational approaches, without the extensive engineering that is needed to obtain state-of-the-art performance on

| 2D Poses | Baseline | L2 | Huber |
|---|---|---|---|
| | 214.2 | 242.4 | 158.9 |

Figure 3: Sample pose from Figure 3 of the main paper, taken from the tail of the error distribution. Even if the reconstruction fails, the reproduced pose is still physically plausible.

Table 1: Quantitative evaluation of the monocular network on the stacked hourglass detections of the Human3.6M dataset, provided by Martinez *et al.* [4]. Baseline results are obtained by initialising the network with the parameter provided by Tome *et al.* [5], while Protocol 1 ($P1$) corresponds to the average per joint 3D reconstruction error, expressed in mm. Protocols 2 and 3 ($P2$ and $P3$) involve an additional step of alignment via Procrustes analysis, and in Protocol 3 only $14$ of the $17$ joints are considered. Results are included for mixtures of 3, 6 and 10 models.

| Monocular | Dir. | Disc. | Eat | Greet | Phone | Photo | Pose | Purch. | Sit | SitD. | Smoke | Wait | WalkD. | Walk | WalkT. | **Avg** |
|---|---|---|---|---|---|---|---|---|---|---|---|---|---|---|---|---|
| Tome *et al.* [5] | 65.0 | 73.5 | 76.8 | 86.4 | 86.3 | 110.7 | 68.9 | 74.8 | 110.2 | 173.9 | 85.0 | 85.8 | 86.3 | 71.4 | 73.1 | 88.4 |
| Martinez *et al.* [4] | 51.8 | 56.2 | 58.1 | 59.0 | 69.5 | 78.4 | 55.2 | 58.1 | 74.0 | 94.6 | 62.3 | 59.1 | 65.1 | 49.5 | 52.4 | 62.9 |
| 3 Models - Baseline | 100.8 | 104.1 | 103.1 | 108.4 | 109.2 | 122.3 | 98.1 | 130.7 | 136.6 | 192.1 | 102.0 | 108.1 | 113.6 | 98.4 | 104.9 | 114.2 |
| 6 Models - Baseline | 100.7 | 91.7 | 103.4 | 108.0 | 123.8 | 118.3 | 98.3 | 91.1 | 137.8 | 207.1 | 104.7 | 97.5 | 103.2 | 94.8 | 103.0 | 112.3 |
| 10 Models - Baseline | 100.7 | 92.2 | 109.6 | 108.3 | 128.2 | 119.1 | 97.8 | 91.1 | 148.7 | 214.6 | 112.0 | 98.1 | 103.5 | 94.8 | 103.0 | 115.4 |
| 3 Models (P1) | 57.5 | 68.5 | 65.9 | 71.3 | 82.0 | 80.2 | 57.5 | 60.7 | 102.8 | 136.7 | 75.8 | 70.6 | 74.4 | 58.2 | 64.4 | 76.1 |
| 6 Models (P1) | 55.0 | 66.6 | 65.0 | 67.5 | 81.5 | 79.6 | 58.1 | 59.1 | 95.0 | 127.3 | 74.7 | 67.6 | 68.3 | 56.6 | 63.4 | 73.4 |
| 10 Models (P1) | 55.8 | 66.3 | 63.2 | 67.9 | 81.7 | 80.5 | 58.9 | 57.8 | 95.0 | 124.6 | 74.4 | 66.0 | 69.3 | 56.5 | 66.0 | 73.2 |
| 3 Models (P2) | 48.4 | 54.0 | 58.4 | 58.2 | 70.2 | 66.2 | 45.2 | 49.9 | 80.9 | 110.3 | 66.1 | 57.0 | 61.0 | 48.5 | 52.8 | 62.7 |
| 6 Models (P2) | 46.4 | 51.1 | 56.3 | 55.0 | 69.9 | 63.7 | 44.7 | 45.6 | 79.1 | 91.3 | 60.9 | 54.2 | 56.9 | 43.7 | 50.3 | 59.0 |
| 10 Models (P2) | 45.6 | 52.1 | 55.1 | 54.9 | 69.0 | 65.5 | 45.3 | 47.6 | 85.7 | 93.6 | 60.4 | 53.6 | 58.5 | 43.4 | 51.1 | 59.7 |
| 3 Models (P3) | 51.0 | 62.3 | 60.0 | 60.3 | 76.2 | 72.3 | 49.0 | 53.5 | 90.4 | 113.3 | 71.2 | 60.8 | 69.7 | 50.4 | 54.6 | 67.6 |
| 6 Models (P3) | 48.7 | 55.0 | 56.2 | 59.2 | 75.4 | 66.6 | 50.1 | 50.4 | 90.5 | 102.8 | 66.4 | 57.9 | 60.8 | 49.8 | 57.6 | 64.2 |
| 10 Models (P3) | 48.0 | 55.8 | 55.3 | 58.3 | 76.3 | 6b9.7 | 48.6 | 48.1 | 88.8 | 100.3 | 66.2 | 56.1 | 59.8 | 48.3 | 57.1 | 63.7 |

the Human3.6M dataset. The results, detailed in Table 1, consider three different Protocols: Protocol 1 coincides with the regular average per-joint error, and is the same protocol used in the main paper. Protocol 2 and 3 instead involve also an alignment of ground truth and reconstruction using Procrustes analysis, before averaging the reconstruction error over $17$ and $14$ joints respectively.

This experiment also allows us to gain further insight in the mixture of models used by Tome *et al.* [5], by looking at one of the trained parameters. In our network, we introduced - and optimised - a series of weights for the $x$ and $y$ components of the reprojection error. We can now plot these weights as heatmaps, placing each weight on the corresponding joint. In Figure 2 we can see the results for the multicamera approach, while Figure 4 displays the same plots but for the three monocular mixtures of models. It is possible to notice how different models are more receptive to specific joints.

## Footnotes

[1]This fact is a consequence of the the positive definiteness of $H_{y^*}$ and the Lipschitz properties of $\nabla_{y^*}$, and $H_{y^*}$ with respect to $w$. Then for small $\Delta$, the second-order Taylor expansion about $y^*$ of $E(y; w + \Delta)$ still has a positive definite Hessian $H_{y^*}(w + \Delta)$, and predicts an isolated minimum close to $y^*$. As $\Delta$ tends to 0 the new minima and $y^*$ get closer and the Taylor approximation becomes a better and better approximation of $E$ over the smallest open disc containing these two points.

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

Figure 4: Learnt importance of joints for the monocular case. The figure shows the relative weights placed by each model on the y (left) and x (right) component of each joint location. As each model represents a specific type of pose, we can see that for a given type of pose certain joints are more significant than others.