[Reviews · NeurIPS 2019]

Reviewer 1



POST-REBUTTAL COMMENTS Thanks for your response. Alternative derivation: Sorry, I wasn't thinking of the alternative derivation (using Newton's method) as a contribution. I still don't see that this derivation gives any additional insight into the problem, however. Partial derivatives: Thanks, I believe some clarifying remarks will make it easier to follow. Deep learning: The main appeal of this paper is that it presents a general mechanism that can be integrated into any gradient-based learning framework. However, the other reviewers have raised legitimate concerns that there are no experiments which demonstrate the approach in an end-to-end setting, and I agree. While you have shown that the stabilised variant works, it has only been demonstrated on relatively shallow problems. Tracking experiments: Putting some frames of a video in the training set and other frames in the testing set violates the independence assumption. To me, this means that you are virtually reporting training error, not testing error. This may be sufficient to show that the optimization procedure works. However, you may as well not bother having a testing set. Related work: Great. The contribution of the paper will be more clear once put in better context. Taking all of this into account, I will preserve my rating of borderline / lean-to-reject. I like the idea of the paper. However, I think that it needs to be further developed before being published. Minor: Perhaps "Kegan et al" should be cited as "Samuel et al"? ORIGINAL REVIEW # Originality The overall method is similar to the prior work of Kegan et al. It seems that the main theoretical difference is the introduction of a trust region regularizer to stabilize the solution? This paper also misses several relevant citations. In particular: * Strelow "General and Nested Wiberg Minimization" (CVPR 2012). This paper extends the Wiberg technique to the optimization of general functions of multiple variables. * Bertinetto et al "Meta-learning with differentiable closed-form solvers" (ICLR 2019). Performs SGD using the derivative of several Newton steps for the solution of a logistic regression problem. * Madry et al. "Towards Deep Learning Models Resistant to Adversarial Attacks" (ICLR 2018). Considers the min-max problem of training a network that is robust to an adversary. It may also make sense to cite some of: * Amos and Kolter "OptNet: differentiable optimization as a layer in neural networks" (ICML 2017). Uses the solution of a Quadratic Program as an intermediate layer in a deep network. * Finn et al "Model-agnostic meta-learning for fast adaptation of deep networks" (ICML 2017). Performs SGD using the derivative of a gradient step with respect to the parameters of a network. * Jenni and Favaro "Deep Bilevel Learning" (ECCV 2018). Solves a nested optimization problem to optimize for the error on a validation mini-batch using several training mini-batches. * Maclaurin et al. "Gradient-based hyperparameter optimization through reversible learning" (ICML 2015). Back-propagates through 100 iterations of SGD to optimize hyper-parameters. * Ionescu et al. "Matrix backpropagation for deep networks with structured layers" (ICCV 2015). Incorporates matrix-valued functions into back-prop, including the solution of an eigenvalue problem (normalized cuts). This paper considers a problem which is different or more general than each of these existing works, although it seems that all of the necessary theoretical elements were already existent in this literature. I like the two problems which are tackled using the proposed method (tracker fusion and 3D pose estimation). This approach is novel for these problems, and they demonstrate its generality. # Quality The main issue for this paper is the lack of theoretical novelty. I'm satisfied with the experimental investigation. # Clarity The paper is mostly clear and easy to understand. I understand why you introduced the final term equal to zero in equation 8 (to arrive at the definition of the partial derivative), but I think it wouldn't hurt to state this explicitly. The division notation of equation 8 is not familiar to me. You are using a / b to denote the matrix a_i / b_j, such that delta_y / delta_x is the Jacobian? It might be worth mentioning. I didn't understand what "frames missing at random" meant in the tracking experiment. # Significance The ideas in this paper are not novel in general. However, there does not seem to be an existing paper that presents these known results in a general framework for making continuous predictions in deep networks. # Comments on derivation I have a minor issue with the derivation in Section 3, but it does not affect the correctness of the final result: I don't think you should equate the RHS of equation 8 to the _total_ derivative d/dw (rather than the partial derivative ∂/∂w) in the RHS of equation 9, since it does not seem to account for the dependence of x* on w? To be precise, I think it is OK to say the RHS is equal to {∂/∂w -HE(x; w)^-1 ∇E(x; w)} evaluated at x = x*(w) but not that the RHS is equal to d/dw -HE(x*; w)^-1 ∇E(x*; w)} since x* depends on w. The end result is the same, except that the final expression in eq 9 will be -HE(x*; w)^-1 {∂/∂w ∇E(x; w)} evaluated at x = x*(w). This is more clear: otherwise it may seem that the derivative d/dw ∇E(x*; w) = 0 since the value ∇E(x*(w); w) = 0 for all w. I guess you were implicitly treating x* as constant when needed and you obtained the correct result. However, this required significantly more effort for me as a reader to understand the mathematics. While trying to understand the derivation in Section 3, I accidentally re-discovered the technique of Kegan et al. Why not use this method to obtain the result of equation 9? It seems much more clear and simple. For example, the entire derivation could be: Let t(w) = ∇E(x*(w); w). Clearly t(w) = 0 for all w and therefore dt(w)/dw = 0. 0 = dt(w)/dw 0 = d/dw ∇E(x*(w); w) 0 = (∂/∂x ∇E(x; w)) dx*/dw + (∂/∂w ∇E(x; w)) dx*/dw = -HE(x*(w); w)^-1 (∂/∂w ∇E(x; w))_{x = x*(w)}

Reviewer 2



Originality: I am not very familiar with the prior work on energy-minimization models, nor am I familiar with the related work the authors mention in section 2, so I can not speak very well to the originality of the authors' proposed approach. From the authors' description, it seems like their approach is related to Keagen et al. which has been made more stable and scalable. Quality: This paper attacks an important problem, provides a unique and useful approach, and demonstrates its utility by achieving SOTA performance on a number of competitive baselines. I believe this to be high quality work. Clarity: The paper was quite clear. The motivation was strong and the math was logically described. To make the problem setup more clear, perhaps the authors could have included a more concrete example of an energy-based optimization problem. This is done in the supplement, but to someone who is less familiar with the vision literature, this problem setup may be a bit foreign and a clear example might be useful. I believe a section on limitations might have been useful. The proposed approach is quite appealing but likely cannot be applied to any bi-level optimization problem. I think it would strengthen the author's claims if they included a section on what kinds of problems they believe their approach can be applied to and what kinds of problems would require further algorithmic development. Specifically, some mention of the computational bottlenecks when scaling to energy-functions with many parameters would be useful in helping a reader understand what types of problems this approach could be applied to. Significance: Energy minimization problems are very common in machine learning and the inability to 'backprop' through their solutions has kept researchers from applying many feature learning techniques to them. This paper presents a method to do just that which could open up a great deal of possibilities adding end-to-end feature learning to these problems. ------------------- post rebuttal -------------------- I feel the author's addressed most of the other reviewer's concerns and mine were somewhat addressed at well. As I enjoyed this paper, I will keep my score the same. But, I am not very familiar with much of the related work discussed so I will also keep my confidence score the same.

Reviewer 3



The idea is of value, however, I think there are few experimental protocols and writing and claims on novelty that can be improved. First, I do not think "Wibergian Learning" is the best name for the approach or the title. Aside from the fact that the method is not immediately obvious from the name, it does not accurately describe what is proposed either. Wibergian approaches (as well describted in [4]) are essentially alternation algorithms, where the fixing of one of the unknowns give rise to a closed-from solution for the other. This work has nothing to do with giving rise to a closed-from solution, it only provides a method to update the parameters of the energy function in a stable manner. It also is not used as an alternation algorithm, since in all the experiments it seems like the inputs to the energy function (that is the output of another set of weights (2D pose detections) or algorithms (trackers)) are all fixed. Furthermore, the technical novelty of the appraoch is limited, since [1] also has previously derived the same update. This paper provides a different derivation, although it follows the same decoupling assumptions, and it does not seem like there is particular value in this new derivation itself (a theory minded reviewer may disagree, however it seems like the techniques are standard). Really the key difference is in adding of the small positive identity matrix to the Hessisan to avoid numerical instability, however this is an age old trick in linear algebra (so much so that finding a reference is challenging). This is the first thing you would do if you run into a numerical problem. This is not to say that that the method proposed in the paper is not valuable. This is to say that the contribution of the method does not lie in the novelty of the algorithm, since it has more or less has been developed before in [1]. Thus I do not think it is appropriate to call the proposed technique novel as this paper does in the introduction (line 41). Given above, in my opinion, the novelty/contribution of the paper lies in how it brings attention to the fact that this type of weight updates energy functions play nicely with the current deep learning based approaches, and would have like to see more experiments and discussions on how limiting the assumptions are in practice, such as the assumption that the local neighborhood of x*(w) is strongly convex and how bad the drift is in practice. It would help to have a toy experiment. Further, the writing may be improved. It's rather confusing that the paper calls the approach "networks" in the human pose estimation experiments, since they are more appropriate to be called as parameters of the energy function (eq 11), and it's not like these parameters are network outputs that take some input. It could make more sense if the stacked hourglass ddetection modules are also being updated at the same time, however this seems to be fixed. The details are also missing, as to how long the training takes, how the weights of the energy function changes over time. Further in the 3D pose experiment, the energy function is made more complex, for example how the individual reprojection losses are now weighted. Although the approach is quite simple, on a first read it is difficult to figure out what exactly is going on. Another contribution of this approach is that the method is quite general. I'm sure there are many dictionary learning papers that have dedicated ways to update the dictionary parameters, but the proposed approach provides a quite a general derivative update, that may be used for any kind of energy functions. While a positive of this approach is that it is quite general, it should still offer analysis as to how competitive it is when it is compared to a parameter update for a specific quadratic energy (for example, without accounting for the reprojection loss weights, if you only updated the basis columns (e_i) for the human pose estimation case, how does the proposed approach compare to doing usual PPCA update to the basis on the Human3.6M training data?) In particular, the experimental protocol should have more ablations. Where is the improvement coming from? In [23] the basis are pretrained on a 3D mocap data. Simply re-tuning the basis on the Human3.6M training data may also give simila rise in the performance. As the paper says, the optimization of several sets of parameters are enforced, such as the model aprameters, the scale and the novel weighting of the camera and the reprojection loss. The improvements from updating these sets of parameters should be shown and be compared to a baseline approach (such as re-training the basis on Human3.6M data). Further, the standard approach that it should compare to or provide some time analysis against is that of cross-validation or grid-search of these weights on the training data. Is the proposed approach faster than simply doing grid-search (for the weights of the reprojection loss for example)? This can be a baseline. One thing that was not clear to me was how the dependance of the weights are being considered. For the human pose experiment, the three sets of parameters, where they updates all together or one by one? What is the difference of these approaches?

[Author Response · NeurIPS 2019]

We thank the reviewers for their extensive comments. We are particularly grateful for the engagement and effort that clearly went into our reviews, and all reviewers seem happy with the substantial potential impact of our approach, and the importance of the problem we're addressing. As R4 pointed out, there are plenty of applications in which energy minimisation was previously successful but has yet to be combined with modern feature learning techniques, because of the limitation in propagating gradients through their solutions. Our method addresses that specific issue, with a framework general enough to be suited for any kind of energy functions, as R6 noted. That said, some reviewers raised significant objections based on a different interpretations of the paper, and we wish to clarify here.

1. **Where is the novelty (R2+R4) / What is the point of the new proofs (R2)?** R2 and R4 are correct in that we're showing that a stabilised variant of [1] works. However, our primary result is to show *why* it works. To this end we: (i) showed that the result of [1] could be re-derived as a fixed point of the Newton method. (ii) Observed that replacing Newton's method with a more stable trust-region based method gave rise to a more stable fixed-point (line 131), and (iii) characterised the relationship between the two updates using an existing result from the optimisation literature (line 141).

2. **Partial derivatives vs full-derivatives (R2).** We believe the confusion here comes from the use of $H_{x*}^{-1}$ to compress notation from line 124 onwards. Every sub-scripted $x^*$, should be understood as a fixed-point arising from the previous use of Newton's method and not as dynamically varying. Given this, partial derivatives and full derivatives coincide. In the final version, the use of partial derivatives in eq. 9, and clarifying remarks at line 119 will make this clear.

3. **'Wiberg optimisation is alternation (see [4]), and an inappropriate description for our work' (R6).** Wiberg methods should be understood as a *replacement* for alternation, whereby some variables are replaced by a function that characterises their minimum value, as a function of the remaining values leading to more informative higher-order gradients and faster convergence. This mischaracterisation by R6 is our fault; we had intended to cite Fitzgibbon's later work [†] which corrected the mischaracterisation of [4], rather than [4] itself.

4. **R6 requested better baselines.** We emphasise that we're modifying a baseline [24] that was published independently as state-of-the art, and as such many obvious tweaks have already been done. In particular: (i) [24], and our baseline, made use of the Human3.6M PPCA basis of [23]; (ii) [24] also performed grid-search over the basic parameters governing the Huber loss, the regularizer coefficient, and the fusion of poses. Much of the benefit to our approach comes from updating the basis shape and regularizer (taken from PPCA) to be consistent with the new joint detections (note the large decrease in performance from [24] to our baseline). We will add an additional baseline to the final version showing this.

5. **Computational bottlenecks and comparisons.** More generally grid-search, although effective, simply does not scale to modern problems defined over large datasets with even moderate numbers of parameters. A single pass over the training set of Human3.6M using the Huber norm takes around 5 hours on a standard CPU. Grid-search using 10 values over k parameters would take $5 \cdot 10^k$ hours. The table below shows the parameters involved in the human pose experiment ($‡$ were optimised with grid-search in [23] and [24]). The tracking experiments had a total of 964 parameters.

| Component | body models | reprojection | cameras | covariance | Huber $‡$ | fusion $‡$ | scaling $‡$ |
|---|---|---|---|---|---|---|---|
| Number of parameters | 3978 | 92 | 36 | 2523 | 9 | 6 | 4 |

In comparison, our stochastic online approach took approximately 120 core hours, for more than $6,500$ parameters, to converge fully (i.e. substantially faster than searching over 2 parameters). All sets of parameters (as R6 was wondering) are trained simultaneously. This is necessary as most parameters are tightly coupled, and some parameters (Huber loss and poses fusion) are tuned on the distribution of the residuals, and their optimal values change as the basis and cameras alter the residuals.

6. **R4 and R6 requested further applications showing tighter integration with deep learning approaches.** This is an area of on going work, requiring substantial engineering worthy of a future paper, and can not be described alongside current work within the page limit.

7. **Additional comments.** All issues raised by the reviewers will be clarified. We thank R2 for pointing out papers we missed in our literature review, and we will include them in the final version of the paper. As R4 suggested, we will also add more examples of energy based optimisation problems (to which our method is applicable). R6 flagged the use of 'network' instead of 'parameters of the energy function' in the pose experiment; we agree the name should be changed. R1 asked about the 'frames missing at random' tracking experiment: each sequence is randomly split between training and testing set, meaning that both sets contain the same sequences but no frame is presented in both. We will change this notation to talk about sampling the training/test sets at the frame and sequence level, and discuss in the text.

[†] J. H. Hong and A. W. Fitzgibbon. Secrets of matrix factorization: Approximations, numerics, manifold optimization and random restarts. In IEEE International Conference on Computer Vision (ICCV), 2015.

[Meta-Review · NeurIPS 2019]

The reviewers agree that there exist some interesting technical details in the paper, but they raise concerns regarding the novelty of the technique and the absence of end-to-end feature learning in some of the experiments. That said, the introduction of the lambda term to regularize the Hessian even though straightforward is likely to lead to more stable meta-learning and non-trivial performance gains. Accordingly I recommend accept as a poster. For the record, I asked for an additional unofficial feedback from another expert in the field and they provided me with the following comments: " I would give it a weak accept. The main contribution seems to be a gradient update wrt to hyperparameters. Although it is very similar to related work, I still think its novel and the addition of the lambda term to regularize the Hessian can potentially be very useful in practice since its common to see wild oscillations in this update (without the lambda term). My only criticism is that the math in Page 4 is a bit hand-wavy. In many scenarios, the author makes approximations that then he takes as the true value. For example, the gradient in Eq. (9) are not the true gradients of x^*, but of its quadratic approximation (assumed in Eq. (5)), but this is presented as the true quantity. If I were reviewing the paper I would ask the authors to introduce a new notation to account for this approximation or include the error bounds in these quantities. I agree with R1 that the paper does not do justice to the literature on hyperparameter optimization. One of the most comprehensive references IMO not mentioned is https://arxiv.org/pdf/1806.04910.pdf "